# A sense of belonging: The role of higher education in retaining quality STEM teachers

**Meena M. Balgopal**[1]*, **Andrea E. Weinberg**[2]*, **Laura B. Sample McMeeking**[3], **Danielle E. Lin Hunter**[1], **Diane S. Wright**[1]

**1** Department of Biology/Graduate Degree Program in Ecology, Colorado State University, Fort Collins, Colorado, United States of America, **2** Mary Lou Fulton Teachers College, Arizona State University, Tempe, Arizona, United States of America, **3** CSU STEM Center, Colorado State University, Fort Collins, Colorado, United States of America

* meena.balgopal@colostate.edu (MMB); andrea.weinberg@asu.edu (AEW)

## Abstract

There is an alarming shortage of qualified STEM teachers in American PK-12 schools. The COVID-19 pandemic may exacerbate this crisis and consequently affect who participates in future STEM innovation. At three points during the pandemic, we surveyed early career teachers who were supported by the National Science Foundation as they began teaching in high-needs school districts. Teachers who felt connected to their professional and academic communities reported intentions to remain in the profession, while those who felt isolated reported intentions of leaving. It is critical for STEM academics to maintain professional relationships with graduates who pursue STEM teaching professions after graduation.

**Data Availability Statement:** The data supporting the results of the current study can be found in the Mountain Scholar Data Repository (http://dx.doi.org/10.25675/10217/232378).

## Introduction

The COVID-19 pandemic has highlighted the need to foster science literacy in the U.S. and has shaken educational systems—and the actors within them—meant to do just that. Responding to crises requires that actors within and across systems demonstrate some level of resilience, enabling the system to withstand both local and more expansive turbulences. The pandemic has made the role schools play in the U.S. educational-economic system more apparent to employers and employees alike [1]. However, for science, technology, engineering, and mathematics (STEM) teachers to thrive, they need resources and a community that reinforces *their sense of belonging as a STEM professional* and subsequently to encourage their own students to pursue STEM studies. This sense of belonging begins in higher education when university STEM faculty members cultivate professional identities but must continue to be reinforced for STEM graduates who pursue PK-12 teaching careers [2–4].

Quality STEM teachers are integral to the success of educational systems, as teachers are the single most important school-related predictor of students' success [5, 6]. Throughout the pandemic, students' physical and emotional well-being, as well as the potential reduction of learning gains [7], have been crucial considerations for teachers, schools, and policy makers alike. However, many teachers feel voiceless and unsupported, especially when decisions are made

**Funding:** This research was supported by a National Foundation grant (2029302) awarded to MMB, AEW, and LBSM. The funders had no role in study design, data collection and analysis, decision to publish, or preparation of the manuscript.

**Competing interests:** The authors have declared that no competing interests exist.

by policymakers without input from educators [8] leaving them to balance increasingly untenable workloads amid concerns about their own physical health, emotional well-being, and financial stability. If STEM teachers leave the profession because of these pressures, the US will be faced with an even greater teacher shortage than what existed before the pandemic [9], especially in high-poverty schools [10], a phenomenon with far-reaching implications for innovation in STEM fields.

The reality is that early career teachers are being asked to keep the system afloat—with little support, few resources, and virtually no input in how they are "used" in the system [11]. Unfortunately, such professional environments are not likely to encourage teachers to take risks and be enterprising, both important for students' future ability to be innovative and entrepreneurial in STEM [12, 13]. Although organizational stress, for some people, motivates them to be problem-solvers, when teachers do not find their workplace to be collaborative or their jobs to be safe, they are less likely to be opportunistic and innovative, all attributes of professional resiliency [14]. Hence, it is critical that the professional resilience of high-quality STEM teachers, those most vulnerable to leaving the teaching profession during a global crisis, is understood [15, 16].

In response to policies designed to decrease the spread of the coronavirus, schools quickly implemented practices to allow for remote schooling, which meant providing teachers with professional development and students with access to technology and internet [17, 18]. Schools and teachers often respond to new policies or curricular changes, so some level of disruption is part of the profession [19]. However, because the scale and scope of the pandemic-related disruptions are beyond what has been previously encountered or studied, our exploratory study was designed to determine teacher perceptions of their experience. Using a clear time of turmoil for the U.S. education system can provide value for improving current and future practices and support structures for teachers [18]. With this and other COVID-era studies, the broader STEM community can make informed decisions of how best to support and advocate for the professionals who can encourage diverse students, including "lost Einsteins," to enter STEM early in their educational journeys [20, 21].

To explore STEM teachers' perceptions of their own professional resilience during the COVID-19 pandemic, we administered a series of three surveys to a sample of National Science Foundation Noyce scholarship recipients, a program designed to recruit STEM majors into teacher education programs. Noyce scholars are awarded funding based on both their high academic achievement and their professional commitment to working in high-needs schools. Our objective was to identify how participants responded to the pandemic as schools shut their doors (Spring 2020), as schools began a new academic year (Fall 2020), and then again as teachers had a chance to respond to new conditions and expectations (late Fall 2020). Thus, this exploratory study aimed to identify what variables affected high-quality and professionally committed STEM teachers' intentions to remain or leave the teaching profession.

## Materials and methods

We contacted Noyce programs at 13 institutions of higher education in six Western and Plains states to assist us in recruiting their Noyce Scholars, including those still in a licensure program but not yet employed as a teacher (preservice) and those currently employed as a teacher (professional). Although our participants were recruited from Noyce programs at universities in the US West, they are now teaching in schools across 18 different states and nearly every participant is teaching in a different district. This research was approved by the Colorado State University institutional review board. All participants provided their electronic consent.

The COVID pandemic provided us an opportunity to survey STEM teachers, but it also required that we create novel instruments to feasibly answer our research question about what variables affect teachers' perceptions of their professional resilience [22]. Our exploratory study followed the norms of crossover analytic studies, which combine qualitative (open-response) and quantitative (closed-response) data, to identify participant perceptions about remaining in the teaching profession [23]. Therefore, we were not testing hypotheses. Surveys included a combination of closed- and open-response items, which varied with each survey, to measure teachers' perceptions of the educational environment within which they were working and how it impacted their personal and professional decisions, especially to remain in or leave the teaching profession [23, 24]. Survey items were informed by publications on teacher experiences during the pandemic, as well as prior research on teacher retention and burnout [7, 8]. Each survey was reviewed by a team of STEM educators who were not a part of our study to ensure content validity, allowing us to make clarifications before administering each survey. In addition, comparison of closed and open responses for each respondent further reinforced the reliability of our surveys. Using content analysis [25], open responses were thematically coded within Dedoose (v8.3.44) based on the emergent themes (e.g., isolation and connectedness), then quantified to conduct inferential analyses including t-tests and one-way ANOVA using SPSS (v27). An *a priori* threshold for significance was set at .05 for quantitative analyses.

Only participants (n = 104) who completed all three surveys were included in this study. We did not do *a priori* sample size or power calculations because our study did not engage in sampling. In other words, we surveyed all Noyce Scholars across the programs. Participants identified as female (n = 66), male (n = 31), gender-non forming (n = 3), or preferred not to respond (n = 4). Most identified as white (n = 87), and a few preferred not to share their race and/or ethnicity (n = 3). Of those who identified as Teachers of Color (TOC), participants described themselves as Asian (n = 2), Black (n = 2), Latine (n = 5), or more than one race/ethnicity (n = 5). Some participants cared for dependents (n = 28; e.g., children, partners, parents, grandparents) while the majority (n = 75) did not or preferred not to respond (n = 1).

Of those who reported what they were teaching, the majority reported science (n = 65), while others taught mathematics (n = 16), engineering (n = 1), or a combination of STEM areas (n = 10). Of the participants teaching, 99% were in secondary (grades 6–12) settings, and 88% were in public schools. Participants included those who had been in the classroom less than a year (n = 26) or more than a year, who we refer to as professional STEM teachers (n = 73), as well as those who moved out of classroom teaching during the study (n = 5). Of the professional teachers, 58% had been teaching fewer than three years, 20% reported teaching between four and five years, and 17% had six or more years of experience.

## Results

Feeling that one belonged to a professional community was important to the preservice and professional STEM teachers surveyed (Fig 1). The relevance of professional communities extended to both teacher perceptions of meeting student needs, as well as meeting their own needs (Fig 1A.1-3). Those who felt connected to a professional community (86%) were more likely than those who felt isolated (14%) to perceive that, in meeting student needs, they received support from networks of teachers from other schools (1**A.1**; $F = 3.88(1,59)$, $p = 0.05$, $\eta^2 = 0.06$), university faculty (**A.2**; $F = 3.09(1,52)$, $p = 0.08$, $\eta^2 = 0.06$), and university resources (1**A.3**; $F = 8.86(1,46)$, $p < 0.01$, $\eta^2 = 0.16$). Moreover, teachers who felt connected to a professional community were more likely than those who felt isolated to perceive that, in meeting their own needs, they received support from networks of teachers from other schools (1**A.1**;

**Fig 1. Pathway to supporting STEM teachers as educational and STEM professionals with the goal of retaining them in STEM education.** Box plots A.1-3 show that external networks, university faculty, and university resources are important factors in teachers' sense of belonging. Box plots B show that professional teachers indicate that they are less likely than preservice teachers to remain in the STEM education profession for the long term. Box plots C shows that teachers of color (TOC) are more likely to identify professional stability, financial stability, and students' educational needs as impacts on their decision to remain in STEM education. The lines depict the potential pathways (i.e., remain or leave) for teachers as they balance supports and barriers, and the levers that might influence these decisions (e.g., professional belonging, resources, stability).

$F = 9.35(1,59)$, $p < 0.01$, $\eta^2 = 0.14$), university faculty (1**A.2**; $F = 5.99(1,50)$, $p = 0.02$, $\eta^2 = 0.11$), and university resources (1**A.3**; $F = 6.77(1,45)$, $p = 0.01$, $\eta^2 = 0.13$). Beyond meeting the needs of students in general, teachers who felt connected were more likely to perceive that they had been successful meeting the needs of students from diverse backgrounds ($F = 5.46(1,74)$, $p = 0.02$, = 0.07). With medium ($\eta^2 \geq .06$) to large ($\eta^2 \geq .14$) effect sizes, data reported were practically significant, offering evidence that connections to STEM professional networks impacts STEM teachers' perceptions of their professional efficacy.

STEM teachers' commitment to continue teaching shifts once they enter the profession (Fig 1B). Professional teachers were less likely to anticipate staying in teaching for 5–10 years (1**B**; $F = 8.91(2,97)$, $p < 0.01$, $\eta^2 = 0.16$) or more than 10 years (1**B**; $F = 8.16(2,97)$, p $< 0.01$, $\eta^2 = 0.13$) compared to preservice teachers. Therefore, STEM teachers' intentions to leave teaching could be mitigated by their perceptions of support they receive from professional networks.

Those who identify as teachers of color (TOC), compared to white teachers, were more likely to explain that professional stability (1**C**; $t = 2.39(98)$, $p = 0.02$, $g = 0.5$) is an important

factor in their decisions to remain as a teacher (Fig 1). Furthermore, financial stability (1**C**; $t = 1.74(98)$, $p = 0.08$, $g = 0.5$), and meeting their students' educational needs (1**C**; $t = 1.78(98)$, $p = 0.08$, $g = 0.6$) are factors that approach significance. While this does not suggest these factors were not relevant to white teachers, it does indicate a heightened relative importance of these variables for TOC. Although the sub-sample is small, this finding points to practical significance and the need for additional research, as others also argued [26, 27].

## Discussion and conclusion

Because STEM teachers in American public schools are required to demonstrate their content expertise by earning a degree in their licensure area, future teachers are enrolled in our undergraduate STEM courses. Respondents in our study indicated that their feelings of being connected to their professional communities, including university professors, helped meet their own and students' needs (e.g., [28]. The professional community to which these STEM teachers belong, therefore, includes, not only their education professors, but their science and mathematics professors [29–31]. Given that isolation can be a key factor in teacher attrition, this sense of belonging noted by the teachers in our study may have affected their intentions to remain in the profession [32, 33]. More research is needed to identify, specifically how and if remaining connected to one's academic community is a mitigating factor in teacher retention. Moreover, intentionally designed studies regarding the experiences of TOCs is paramount to understanding how feelings of professional belonging, along with professional and financial stability, may help retain PK-12 TOC, where they can be important role models for future STEM professionals.

Our survey findings underscore the importance of STEM teachers feeling supported by their professional networks, which include their alma maters. In addition, our findings reiterate what others have reported regarding the need to pay STEM teachers a competitive salary [34, 35]. Some teachers' decisions to remain in teaching are influenced by their desire for both financial and professional stability, highlighting the need to pay STEM teachers the salaries that are commensurate with other STEM professions. We acknowledge limitations of our study (e.g., participants only included NSF scholarship recipients who graduated from institutions in a limited geographic range, not all respondents completed all three surveys, focus group interview analyses are not included in this study). Nonetheless, our results are informative. Teachers benefit from supportive networks and resources to meet their students' needs, as well as their own. Hence, when universities and school districts partner, there are opportunities to retain high-quality STEM teachers who benefit from feeling valued [36].

Finally, universities and school districts should collaborate to provide professional development training to build individuals' strengths, increase their self-efficacy, and address psychological and professional resilience [37, 38]. Professionally resilient STEM teachers are better equipped to model for students how to see these turbulences as opportunities and how to discover their own adaptive capacities [39]. We are currently examining the relationship between teachers' sense of professional optimism and their intended resilience, as well as characterizing what specific policies evoked feelings of professional stress. Future studies on STEM teacher resilience should also examine the nuanced relationship between teacher beliefs and student self-efficacy, achievement, and interest in STEM disciplines.

Teaching STEM goes far beyond the transmission of knowledge. Ultimately, it is about understanding how knowledge is generated (i.e., scientific processes and innovation) to develop both future STEM professionals and STEM-literate citizens. Those who value social justice should also value the role that PK-12 STEM teachers play in ensuring success across diverse communities. Therefore, to ensure that high-quality STEM teachers have the best

opportunities for success, we encourage our academic colleagues in natural sciences to collaborate with teacher education programs and school districts to promote teacher resilience. Such investments can benefit STEM literacy of all K-12 graduates.

## Author Contributions

**Conceptualization:** Meena M. Balgopal, Andrea E. Weinberg, Laura B. Sample McMeeking, Diane S. Wright.

**Data curation:** Danielle E. Lin Hunter, Diane S. Wright.

**Formal analysis:** Danielle E. Lin Hunter, Diane S. Wright.

**Funding acquisition:** Meena M. Balgopal, Andrea E. Weinberg, Laura B. Sample McMeeking.

**Project administration:** Meena M. Balgopal.

**Visualization:** Laura B. Sample McMeeking, Danielle E. Lin Hunter.

**Writing – original draft:** Meena M. Balgopal.

**Writing – review & editing:** Meena M. Balgopal, Andrea E. Weinberg, Laura B. Sample McMeeking, Danielle E. Lin Hunter, Diane S. Wright.

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
