## [Decision Letter · Decision Letter 0]

21 Oct 2021

PONE-D-21-18404A sense of belonging: The role of higher education in retaining quality STEM teachersPLOS ONE

Dear Dr. Balgopal,

Thank you for submitting your manuscript to PLOS ONE. After careful consideration, we feel that it has merit but does not fully meet PLOS ONE’s publication criteria as it currently stands. Therefore, we invite you to submit a revised version of the manuscript that addresses the points raised during the review process.

We look forward to receiving your revised manuscript.

Kind regards,

Fraide Agustin Ganotice, PhD

Academic Editor

PLOS ONE

Journal Requirements:

Additional Editor Comments (if provided):

Dear Dr. Meena M. Balgopal,

Thank you for submitting your manuscript titled “A sense of belonging: The role of higher education in retaining quality STEM teachers” to PLOS ONE. Four reviewers have examined the manuscript for which one recommended rejection, one recommended minor revision and two recommended major revision. I also went over the manuscript and agreed with the two reviewers to recommend major revision. The manuscript, though important, needs to be rewritten and to strictly follow the comments made by the reviewers. I encourage you to study the comments made and revise your manuscript following the comments made.

Thank you so much!

Best regards,

Fred Ganotice

Reviewers' comments:

Reviewer's Responses to Questions

**Comments to the Author**

1. Is the manuscript technically sound, and do the data support the conclusions?

Reviewer #1: No

Reviewer #2: No

Reviewer #3: Yes

Reviewer #4: Partly

2. Has the statistical analysis been performed appropriately and rigorously? 

Reviewer #1: No

Reviewer #2: No

Reviewer #3: Yes

Reviewer #4: I Don't Know

3. Have the authors made all data underlying the findings in their manuscript fully available?

Reviewer #1: Yes

Reviewer #2: Yes

Reviewer #3: Yes

Reviewer #4: No

4. Is the manuscript presented in an intelligible fashion and written in standard English?

Reviewer #1: Yes

Reviewer #2: Yes

Reviewer #3: Yes

Reviewer #4: Yes

5. Review Comments to the Author

Reviewer #1: I have read the whole article and made a decision that this article will not be considered further. Even this is a well-written article, the descriptive analysis used is not rigorous enough on exploring the issue under studied.

Reviewer #2: 1. The findings are not directly tied to the data analysis and there seems to be some bias present in the findings.

2. The paper says data were analyzed using SPSS, but does not specify what kind of statistical analysis was done. Although the data says qualitative data was collected and analyzed, no qualitative data was presented.

3. The results section was difficult to follow, it took several reads to understand what was being presented. I recommend explaining what the 1A, 1B etc. in the body of the paper.

Reviewer #3: Line 46 - "as decision are made" is vague. By whom?

Lines 82-88 - some information on how data were aggregated is needed here. What were emerging themes, if any?

Figure 1: y-axes should be labeled.

Discussion: some mention of design limitations is needed (specifically geographic limitations). Otherwise, I think the authors are overgeneralizing their findings. Ideas for future research would be a nice addition to the discussion. Some information on regional demographics would also be good to characterize the economic state of where the data were collected.

Reviewer #4: In order to evaluate this manuscript fully the following information is required:

1. more detail in the methods, especially about data analysis

2. more details about participant demographics, including age, gender, race/ethnicity, etc

6. PLOS authors have the option to publish the peer review history of their article (what does this mean?). If published, this will include your full peer review and any attached files.

Reviewer #1: No

Reviewer #2: No

Reviewer #3: No

Reviewer #4: **Yes: **Denise Adams

---

## [Author Response · Author response to Decision Letter 0]

9 Nov 2021

Please see attached document (Response to Reviewers) for a detailed description of how we responded to edits.

---

## [Decision Letter · Decision Letter 1]

15 Feb 2022

PONE-D-21-18404R1

A sense of belonging: The role of higher education in retaining quality STEM teachers

PLOS ONE

Dear Dr. Balgopal,

Thank you for submitting your manuscript to PLOS ONE. After careful consideration, we feel that it has merit but does not fully meet PLOS ONE’s publication criteria as it currently stands. Therefore, we invite you to submit a revised version of the manuscript that addresses the points raised during the review process.

-The study is suggesting commitment to school and profession indicating either normative, affective, continuance commitment (Meyer and Allen, 1997) framework and scale. This manuscript is not explicit about the validated scale used to collect the data.

-You mentioned three surveys were administered - there is a need to report the validation procedure on the scale - any data collected from non-valid measures are not acceptable,

-The introduction is also mentioning resilience in addition to commitment, which is another variable with other standard scale being used

-The objectives should suggest the methodology

-Page 5 suggest that you used mixed methods - qualitative and quantitative but you need to report the scale - the validation procedure of the scale

-I suggest to have your data done by professional statistician and a qualitative researcher in order to improve the presentation. For example there is no mention what specific problems in which you used t-test and ANOVA useful? What are the mean scores of the variables being compared here.nd not, for example, on novelty or perceived impact.

We look forward to receiving your revised manuscript.

Kind regards,

Fraide Agustin Ganotice, PhD

Academic Editor

PLOS ONE

Journal Requirements:

Additional Editor Comments:

Dear Prof Chenette,

Thank you for resubmitting your manuscript titled "A sense of belonging..." for consideration of PLOS One. I invite you to pay attention to the additional comments and suggestions given by the reviewers and resubmit for another round of review.

I also gave my comments below:

-The study is suggesting commitment to school and profession indicating either normative, affective, continuance commitment (Meyer and Allen, 1997) framework and scale. This manuscript is not explicit about the validated scale used to collect the data.

-You mentioned three surveys were administered - there is a need to report the validation procedure on the scale - any data collected from non-valid measures are not acceptable,

-The introduction is also mentioning resilience in addition to commitment, which is another variable with other standard scale being used

-The objectives should suggest the methodology

-Page 5 suggest that you used mixed methods - qualitative and quantitative but you need to report the scale - the validation procedure of the scale

-I suggest to have your data done by professional statistician and a qualitative researcher in order to improve the presentation. For example there is no mention what specific problems in which you used t-test and ANOVA useful? What are the mean scores of the variables being compared here.

Please go over the reviewers' comments. You can also make reference to other completed and published good articles for you to have a mental frame of a publishable paper. I agree that this paper is good but there is a need to overhaul the writing of the manuscript. hope you will find these comments helpful as you further improve your work.

Best wishes,

Fred Ganotice

Reviewers' comments:

Reviewer's Responses to Questions

**Comments to the Author**

1. If the authors have adequately addressed your comments raised in a previous round of review and you feel that this manuscript is now acceptable for publication, you may indicate that here to bypass the “Comments to the Author” section, enter your conflict of interest statement in the “Confidential to Editor” section, and submit your "Accept" recommendation.

Reviewer #2: (No Response)

Reviewer #3: All comments have been addressed

Reviewer #4: All comments have been addressed

2. Is the manuscript technically sound, and do the data support the conclusions?

Reviewer #2: Partly

Reviewer #3: Yes

Reviewer #4: Yes

3. Has the statistical analysis been performed appropriately and rigorously? 

Reviewer #2: Yes

Reviewer #3: Yes

Reviewer #4: Yes

4. Have the authors made all data underlying the findings in their manuscript fully available?

Reviewer #2: Yes

Reviewer #3: Yes

Reviewer #4: Yes

5. Is the manuscript presented in an intelligible fashion and written in standard English?

Reviewer #2: Yes

Reviewer #3: Yes

Reviewer #4: Yes

6. Review Comments to the Author

Reviewer #2: Figures:

You state in your figures that Pre-pandemic STEM graduates entering the teaching profession are supported by higher education and STEM net works. You need citations to support this claim.

Figure 1A rather than saying "some teachers" include the N or percentage.

Figure 1B says professional STEM teachers were less likely than preservice to remain, but your survey asked about intentions, so it doesn't seem you have the data to support this claim.

In the figure it says "A sense of isolation or belonging could impact teacher's..." this seems like opinion rather than data from your study.

The N for teachers of color is really small, so I recommend not reporting findings on those statistics.

Discussion & Conclusion:

I don't see how the data supports the statement: "Ultimately, many of today’s PK-12 students are future STEM professionals.

Because STEM teachers in American public schools are required to demonstrate their

content expertise by earning a degree in their licensure area, it is likely that many of

our undergraduate STEM students are future teachers."

I don't see how the data supports the statement: "Our findings indicate that the

importance of how connected STEM teachers feel to their professional communities,

including university professors, may predict their intentions to remain in the

profession." Additionally, "may predict" does not inform any actionable interventions.

Again - the N for TOC is really small, so I recommend leaving these results out.

"Some teachers’ decisions to remain in teaching are influenced by their desire for both financial and professional stability, highlighting the need to pay STEM teachers salaries commensurate with other STEM

professions." this finding was only for TOC correct?

Reviewer #3: (No Response)

Reviewer #4: (No Response)

7. PLOS authors have the option to publish the peer review history of their article (what does this mean?). If published, this will include your full peer review and any attached files.

Reviewer #2: No

Reviewer #3: No

Reviewer #4: No

---

## [Author Response · Author response to Decision Letter 1]

12 Apr 2022

We have uploaded a detailed matrix entitled Response to Reviewer Comments.

---

## [Decision Letter · Decision Letter 2]

30 Jun 2022

PONE-D-21-18404R2

A sense of belonging: The role of higher education in retaining quality STEM teachers

PLOS ONE

Dear Dr. Balgopal, (cc: Academic Editor Dr. Fred Ganotice)

Thank you for submitting your manuscript to PLOS ONE. After careful consideration, we feel that it has merit but does not fully meet PLOS ONE’s publication criteria as it currently stands. Therefore, we invite you to submit a revised version of the manuscript that addresses the points raised during the review process.

Please revise the article according to the reviewer's comment, especially the effect size description at the top of page 8.

We look forward to receiving your revised manuscript.

Kind regards,

Yann Benetreau

Staff Editor

PLOS ONE

Journal Requirements:

Reviewers' comments:

Reviewer's Responses to Questions

**Comments to the Author**

1. If the authors have adequately addressed your comments raised in a previous round of review and you feel that this manuscript is now acceptable for publication, you may indicate that here to bypass the “Comments to the Author” section, enter your conflict of interest statement in the “Confidential to Editor” section, and submit your "Accept" recommendation.

Reviewer #2: (No Response)

Reviewer #3: (No Response)

Reviewer #4: All comments have been addressed

2. Is the manuscript technically sound, and do the data support the conclusions?

Reviewer #2: No

Reviewer #3: Partly

Reviewer #4: Yes

3. Has the statistical analysis been performed appropriately and rigorously? 

Reviewer #2: No

Reviewer #3: No

Reviewer #4: Yes

4. Have the authors made all data underlying the findings in their manuscript fully available?

Reviewer #2: Yes

Reviewer #3: Yes

Reviewer #4: Yes

5. Is the manuscript presented in an intelligible fashion and written in standard English?

Reviewer #2: Yes

Reviewer #3: Yes

Reviewer #4: Yes

6. Review Comments to the Author

Reviewer #2: At the top of page 8, the effect size is listed as: With medium (≥.06) to large (≥.14) effect sizes. However, an effect size of.06 is not medium, its small. Similarly an effect size of.14 is not large, its small. I also pulled the original data from the website and I do not see a question asking if they feel they belong to a professional community. Did the authors combine responses to a set of questions and roll that up to be connected to a professional community?

Reviewer #3: Your a priori alpha level needs to be stated in the methods section. Choosing p=0.08 as marginally significant just because it has a medium effect size is not acceptable. There is really no such thing as marginal significance. It either is or isn't. That's why we have effect size indicators. The words "approached significance" would be acceptable, unless your a priori alpha is 0.08, which would need to be justified as it is non-standard.

Reviewer #4: This revision has adequately addressed previous concerns of all reviewers and I recommend proceeding with publication of this version.

7. PLOS authors have the option to publish the peer review history of their article (what does this mean?). If published, this will include your full peer review and any attached files.

Reviewer #2: No

Reviewer #3: No

Reviewer #4: No

---

## [Author Response · Author response to Decision Letter 2]

30 Jun 2022

Reviewer 2 Comment 1: At the top of page 8, the effect size is listed as: With medium (≥.06) to large (≥.14) effect sizes. However, an effect size of.06 is not medium, its small. Similarly an effect size of.14 is not large, its small. 

RESPONSE/REVISION: A partial eta squared was used as an effect size measure for our ANOVA analyses. The reviewer offers the interpretations for cohen's d (small (d = 0.2), medium (d = 0.5), and large (d = 0.8), Cohen, 1988). However, our analysis and interpretation rely on a partial eta squared, with benchmarks of small (η2 = 0.01), medium (η2 = 0.06), and large (η2 = 0.14) effects (Cohen, 1988; Olejenik and Algina, 2003). We notice that while we include the η2 statistic with the ANOVA results, we did not include the statistical symbol "η2" with the interpretation of effect size. We have now included this for clarification.

Reviewer 2 Comment 2: I also pulled the original data from the website and I do not see a question asking if they feel they belong to a professional community. Did the authors combine responses to a set of questions and roll that up to be connected to a professional community?

RESPONSE: Connectedness and its opposite, isolation, were strong themes that emerged during the the thematic coding of open-response items (see manuscript p. 6). Various types of connectedness emerged (i.e., professional, familial, social). Given the focus of this paper on professional belonging, we quantified based on this emergent theme of "professional connectedness" and made comparisons between those who described feelings of connectedness to those who did not. Our analysis indicated that those who felt professionally connected (as indicated in the open-ended responses) were more likely than those who felt isolated to perceive they received support in meeting student needs from networks of teachers from other schools, university faculty, and university resources (as reported on Likert-type items).

Reviewer 3 Comment 1: Your a priori alpha level needs to be stated in the methods section.

RESPONSE/REVISION: The a priori alpha level is now stated in the methods section, following our explanation of the statistical analyses themselves.

Reviewer 3 Comment 2: Choosing p=0.08 as marginally significant just because it has a medium effect size is not acceptable. There is really no such thing as marginal significance. It either is or isn't. That's why we have effect size indicators. The words "approached significance" would be acceptable, unless your a priori alpha is 0.08, which would need to be justified as it is non-standard.

RESPONSE/REVISION: The section discussing these findings has been revised. We no longer describe the factors that have a .08 significance level as "marginally significant". We have used the reviewer's suggested language of "approached significance."

---

## [Decision Letter · Decision Letter 3]

22 Jul 2022

A sense of belonging: The role of higher education in retaining quality STEM teachers

PONE-D-21-18404R3

Dear Dr. Balgopal,

We’re pleased to inform you that your manuscript has been judged scientifically suitable for publication and will be formally accepted for publication once it meets all outstanding technical requirements.

Kind regards,

Jianhong Zhou

Staff Editor

PLOS ONE

Additional Staff Editor Comments: Please include additional information regarding the surveys used in the study and ensure that you have provided sufficient details that others could replicate the analyses. For instance, if you developed the surveys as part of this study and they are not under a copyright more restrictive than CC-BY, please include a copy as Supporting Information.

In addition, please include a sample size and power calculation in the Methods, or discuss the reasons for not performing one before study initiation.

Reviewers' comments:

Reviewer's Responses to Questions

**Comments to the Author**

1. If the authors have adequately addressed your comments raised in a previous round of review and you feel that this manuscript is now acceptable for publication, you may indicate that here to bypass the “Comments to the Author” section, enter your conflict of interest statement in the “Confidential to Editor” section, and submit your "Accept" recommendation.

Reviewer #2: All comments have been addressed

Reviewer #3: All comments have been addressed

Reviewer #4: All comments have been addressed

2. Is the manuscript technically sound, and do the data support the conclusions?

Reviewer #2: (No Response)

Reviewer #3: Yes

Reviewer #4: Yes

3. Has the statistical analysis been performed appropriately and rigorously? 

Reviewer #2: (No Response)

Reviewer #3: Yes

Reviewer #4: Yes

4. Have the authors made all data underlying the findings in their manuscript fully available?

Reviewer #2: (No Response)

Reviewer #3: Yes

Reviewer #4: Yes

5. Is the manuscript presented in an intelligible fashion and written in standard English?

Reviewer #2: (No Response)

Reviewer #3: Yes

Reviewer #4: Yes

6. Review Comments to the Author

Reviewer #2: (No Response)

Reviewer #3: (No Response)

Reviewer #4: (No Response)

7. PLOS authors have the option to publish the peer review history of their article (what does this mean?). If published, this will include your full peer review and any attached files.

Reviewer #2: No

Reviewer #3: No

Reviewer #4: No

---

## [Editor Report · Acceptance letter]

8 Aug 2022

PONE-D-21-18404R3 

A Sense of Belonging: The Role of Higher Education in Retaining Quality STEM Teachers 

Dear Dr. Balgopal:

I'm pleased to inform you that your manuscript has been deemed suitable for publication in PLOS ONE. Congratulations! Your manuscript is now with our production department. 

Kind regards, 

on behalf of

Dr. Yann Benetreau 

Staff Editor

PLOS ONE